# Hypoxia Pathway in Osteoporosis: Laboratory Data for Clinical Prospects

**DOI:** 10.3390/ijerph20043129

**Published:** 2023-02-10

**Authors:** Jianping Wang, Bin Zhao, Jingmin Che, Peng Shang

**Affiliations:** 1School of Life Sciences, Northwestern Polytechnical University, Xi’an 710072, China; 2Key Laboratory for Space Bioscience and Biotechnology, Institute of Special Environmental Biophysics, School of Life Sciences, Northwestern Polytechnical University, Xi’an 710072, China; 3Research & Development Institute in Shenzhen, Northwestern Polytechnical University, Shenzhen 518057, China

**Keywords:** hypoxia pathway, osteoporosis, bone remodeling, iron metabolism

## Abstract

The hypoxia pathway not only regulates the organism to adapt to the special environment, such as short-term hypoxia in the plateau under normal physiological conditions, but also plays an important role in the occurrence and development of various diseases such as cancer, cardiovascular diseases, osteoporosis. Bone, as a special organ of the body, is in a relatively low oxygen environment, in which the expression of hypoxia-inducible factor (HIF)-related molecules maintains the necessary conditions for bone development. Osteoporosis disease with iron overload endangers individuals, families and society, and bone homeostasis disorder is linked to some extent with hypoxia pathway abnormality, so it is urgent to clarify the hypoxia pathway in osteoporosis to guide clinical medication efficiently. Based on this background, using the keywords “hypoxia/HIF, osteoporosis, osteoblasts, osteoclasts, osteocytes, iron/iron metabolism”, a matching search was carried out through the Pubmed and Web Of Science databases, then the papers related to this review were screened, summarized and sorted. This review summarizes the relationship and regulation between the hypoxia pathway and osteoporosis (also including osteoblasts, osteoclasts, osteocytes) by arranging the references on the latest research progress, introduces briefly the application of hyperbaric oxygen therapy in osteoporosis symptoms (mechanical stimulation induces skeletal response to hypoxic signal activation), hypoxic-related drugs used in iron accumulation/osteoporosis model study, and also puts forward the prospects of future research.

## 1. Introduction 

As one of the basic components of the atmosphere, oxygen (O_2_) is essential for normal life activities. It primarily affects the structure and activity of DNA and protein biomolecules, and also plays an essential part in the maintenance of life, such as in glycolysis [1,2]. Moreover, O_2_ regulation in the special environment of high altitude is vital [3]. In 2019, the three scientists William G. Kaelin Jr (US), Gregg L. Semenza (US) and Sir Peter J. Ratcliffe (UK) were awarded the Nobel Prize in Physiology or Medicine for their great discovery of how cells perceive and adapt to O_2_ supply [4]. The hypoxia pathway not only provides a new strategy for renal anemia and renal clear cell carcinoma [5], but for other disease models [6,7,8]. 

Under normal physiological conditions, PaO_2_ is about 80–100 mmHg; PvO_2_ is about 40 mmHg [9]. Poussa [10] et al. suggested that the O_2_ partial pressure of the periosteum had an important effect on its differentiation. Under high O_2_ pressure, the periosteal cells mainly differentiated into osteoblasts and generated bone, while under low O_2_ pressure, the periosteal cells mainly differentiated into chondrocytes to generate cartilage.

Bone remodeling is jointly maintained by osteoblasts, osteoclasts and the microenvironment. When the bone resorption function of osteoclasts induced is greater than the bone formation function of osteoblasts induced, bone remodeling is unbalanced, leading to the occurrence of osteoporosis. As is known to all, osteoporosis disease has become a common serious disease that threatens health. China carried out the first epidemiological survey of osteoporosis among Chinese residents in 2018. The survey showed that the prevalence of osteoporosis in over 50-year-olds was 19.2%. More recent studies have shown that the hypoxia pathway is critical in osteoporosis disease. Recently, it has been confirmed that bone regeneration in osteoporosis is the result of the activation of the HIF pathway leading to the expression of the downstream gene *VEGF* [11]. The BMP signal is responsive to mTOR, HIF, Wnt and other key signaling pathways to coordinate energy metabolism and bone homeostasis [12]. The expression of miRNA affects the osteogenic differentiation of BMSCs. Studies have shown that the inhibition of miR-139-5p significantly promotes hBMSC osteogenesis. The Wnt/β-catenin pathway might directly target key factors CTNNB1 and Crimp 4 (FZD4) involved in osteogenesis [13]. miRNA also regulates HIF signaling to a certain extent. Serocki, M. et al. reported that miR-429, miR-155 and miR-200b regulated acute hypoxia signaling. miR-429 and miR-210 played important regulatory roles in chronic hypoxia [14].

There are also studies on osteoporosis/fracture treatment using the hyperbaric oxygen chamber, and the use of external mechanical stimulation or hypoxic-related drugs to activate/mobilize the hypoxia pathway to improve the symptoms of osteoporosis; both in vivo and in vitro experiments have studied and applied these methods. To sum up, the discovery of the hypoxia pathway has created a new research direction, which has meaningful theoretical value and great application potential. It is significant to introduce hypoxia, and the relationship between the hypoxia pathway and osteoporosis disease specifically from the following perspectives to lay a good foundation for further clinical research.

## 2. Osteoporosis

Osteoporosis (OP) is a kind of bone metabolic disease associated with iron overload, which is characterized by low bone mass and the destruction of the bone microstructure, thus resulting in an increase in bone fragility [15]. Half of women over 50 will experience osteoporotic fracture in their lifetime [16]. Also, astronauts suffer from serious bone loss after space flight. After one month of space flight, the bone density of the femur, spine and bone decreased by 6–8% [17]. Osteoporosis has become a major disease affecting human life and health, which can be divided into primary, secondary and idiopathic osteoporosis according to etiology. The etiology of primary OP is complicated, and the pathogenesis is diversified, such as sexual hormone secretion disorder, calcium absorption anomaly, lack of exercise, puberty growth spurt and aberrant bone mass increase, which can all disturb physiological bone homeostasis, leading to varying degrees of OP. Secondary OP is caused by some drugs or diseases, such as glucocorticoids [18], diabetes [19,20], epilepsy [21]; secondary OP is also caused by reduced bone mechanical tension stimulation [22], a study finding that four astronauts in space for 5 to 7 months experienced vertebral bone density loss of 6.1%, 0.3%, 2.3%, 10.8%, respectively [23,24].

The pathogenesis of OP is complex, and underlying mechanisms are diverse. Treatment can be divided into drug therapy, physical therapy and other measures. Drugs include antiresorptive drugs, osteogenesis drugs and other researched drugs or small molecular compounds such as bisphosphonates and their derivatives, cathepsin K inhibitors, calcitonin, estrogen, selective estrogen receptor modulators [25,26,27], calcium-sensitive receptor antagonists, the sclerosing protein antibody EVENITY, Wnt signal agonists and antibody therapy, endogenous Wnt signal inhibitors, parathyroid hormone analogues [28,29,30,31], metformin, vitamin K2, strontium salt, active VD and its analogues and so on [32,33,34,35]. Physical therapies include mechanical stimulation, damp-heat therapy, electromagnetic therapy and static magnetic field therapy [17,36,37,38,39,40,41,42,43,44,45,46,47,48]. MicroRNAs (miRNAs), small interfering RNAs (siRNAs), long non-coding RNAs (lncRNAs), circular RNAs (circRNAs), PIWI-interacting RNAs (piRNAs) and other molecules also play an important role in bone development [49,50]. miRNAs are a new group of small, noncoding RNAs of 19–25 nucleotides; they negatively regulate gene expression after transcription, which has an essential role in bone homeostasis as a regulatory factor by targeting mRNA with partially complementary sequences. miRNA dysregulation is associated with the pathogenesis of cardiovascular diseases, metabolic syndromes and degenerative diseases. A large number of miRNAs have been found to participate in bone homeostasis. Bellavia, D. et al. reviewed the role of miRNAs in the normal and abnormal growth and development of bone, explaining that miRNAs regulate the process of bone reconstruction jointly mediated by osteoblasts and osteoclasts and also maintain bone homeostasis during physiological processes. In pathological conditions, the disorder of the miRNA signaling pathway promotes the occurrence and development of osteoporosis disease [51,52,53,54,55]. The expression level of miR 21a-5p, miR-27a-3p, miR -145, miR-22-3p, miR-340-5p and miR-23a-5p is closely related to the biological behaviors of osteoclasts, such as proliferation, differentiation and apoptosis. miR-222-3p and miR-148a-3p play a role in the signaling pathway network during osteogenic differentiation [56,57,58,59]. Therefore, monitoring the expression level of miRNA related to bone homeostasis can be one of the diagnostic methods for bone diseases, and its regulation can become a new means of the prevention and treatment of osteoporosis diseases. miRNAs show great promise as biomarkers and potential therapeutic targets for osteoporosis.

While there are still controversies in clinical screening methods and diagnostic thresholds for OP, and the treatment of OP is not standardized, especially under the background of graded diagnosis and treatment, it is necessary to strengthen the prevention and treatment ability of general practitioners, and more mechanisms need to be clarified and explained [40,41].

## 3. Bone Remodeling

Bone is in a dynamic balance between bone formation mediated by osteoblasts and bone resorption mediated by osteoclasts, which is called bone remodeling [42].

Osteoblasts are specialized mesenchymal cells that synthesize the bone matrix and coordinate bone mineralization. These cells work in coordination with osteoclasts, which absorb bone, and continue to circulate throughout the life cycle. Bone remodeling consists of four stages: bone resorption, the reversal of bone resorption to bone formation, bone formation and mineralization. During the bone remodeling cycle, bone resorption is tightly coupled to bone formation. The remodeling cycle occurs within the basic multicellular unit and comprises five coordinated steps: activation, resorption, reversal, formation and termination. Osteoblasts produce a range of different secretory molecules, including M-CSF, RANKL/OPG, WNT5A and WNT16, which promote or suppress osteoclast differentiation and development. Osteoclasts also influence osteoblast formation and differentiation through the secretion of soluble factors, including S1P, SEMA4D, CTHRC1 and C3 [60]. 

The process of bone remodeling is influenced and regulated by many factors, such as growth factors, hormones, cytokines, electrokinetic stimulation [17,43,44] and other physical factors [45]. Biochemical molecules are mainly regulated by growth factors, hormones and cytokines secreted by the bone matrix. S1P acts as an osteoclast-osteoblast coupling factor to promote osteoblast proliferation and bone formation. Moreover, the recruitment of osteoclast precursors to resorption sites is regulated by the interplay of S1P gradients and S1P receptor expression [61]. Mechanical forces are indispensable for bone homeostasis; the loss of mechanical stimulation can significantly weaken the bone structure, causing osteoporosis and increasing the risk of fracture [62]. Mechanical stimulus regulation refers to the fact that mechanical load increases bone remodeling [44]. Other physical factors mainly refer to electromagnetic fields of various forms and intensification, which all have certain effects on bone remodeling [46,47]. Static magnetic fields can prevent and treat OP, and can also promote fracture healing and bone regeneration [48]. When the bone remodeling balance is disrupted, bone formation is reduced and bone resorption increased, which leads to the occurrence and development of osteoporosis; that is to say, after the occurrence of bone destruction, the body promotes bone remodeling, and bone remodeling will lead to reversible/irreversible bone loss. Reversible bone loss represents bone resorption, bone refilling and new bone mineralization in sequence. After repair, bone mass and bone are both restored. However, in irreversible bone loss, there occurs an osteoclast-osteogenic imbalance. Although certain new bone formation occurs, the newly born bone is poor, and bone structure destruction easily occurs again, leading to the occurrence of osteoporosis [63,64]. WNT/β-catenin and TGF-BMP signaling are all involved in the process; the hypoxia pathway is one of the mechanisms in osteoporosis development and progression [65]. Therefore, the imbalance of bone remodeling caused by various physiological/pathological and physical/chemical factors is the biological basis for diseases such as osteoporosis. The intervention of bone remodeling in the process of osteoporosis can reverse the state of osteoporosis to some extent and improve the prognosis.

## 4. Hypoxia

### 4.1. Discovery and Introduction of Hypoxia-Related Molecules

At the end of the 20th century, Professor Gregg L. Semenza found that the organism’s adaptive response to hypoxia could increase the content of erythropoietin (EPO) [66]. After that, the protein HIF-1 was isolated and purified [67]. Later, William G. Kaelin Jr showed that the mutation of the Von Hippel–Lindau (*VHL*) gene destroyed the degradation ability of HIF-1α in normoxia, indicating that VHL protein was actually the key subunit of the ubiquitin ligase (E3) complex [68], and Sir Peter J. Ratcliffe further proved that VHL protein could interact with HIF-1α to promote its ubiquitination and degradation under a hypoxia environment [69]. William G. Kaelin Jr and Sir Peter J. Ratcliffe further proved that the proline of HIF-1α had hydroxylation modification. When hydroxylation was blocked, the HIF-1α reaction was destroyed by VHL recognition and ubiquitination subsequently, and the protein stability increased [70]. Later, Sir Peter J. Ratcliffe and his colleagues identified a dioxygenase from Drosophila, which was responsible for catalyzing the hydroxylation of HIF-1α [71]. The mammalian dioxygenase is called the prolyl hydroxylase domain (PHD), which is composed of three types, named PHD1, PHD2 and PHD3 [72]. In order to make a function of PHD, O_2_, α-ketoglutarate, Fe^2+^ and vitamin C (VC) are required [73]. In addition, another hydroxylase named the HIF inhibiting factor (FIH) has been identified. Its catalytic effect is to prevent binding with transcriptional coactivating proteins such as histone acetyltransferase P300, resulting in a reduction in transcriptional activation [74]. The role of two hydroxylases strictly makes the organism abide by the regulation of O_2_. PHD1, PHD2 and PHD3 all have their own functions [75].

HIF-1 contains two subunits: HIF-1α and HIF-1β. HIF-1β is the same as the ARNT, which is an indispensable structural component of the HIF-1 protein. HIF-1α is an indispensable functional component of HIF-1, and its protein content is strictly regulated by O_2_ concentration [76]. Now, it has been found that there are three HIF-α subunits (HIF-1α, HIF-2α and HIF-3α) [52] as well as three HIF-β subunits (ARNT1, ARNT2, ARNT3) [77,78,79], which can form a variety of heterodimer response mechanisms to hypoxia. HIF-1α is widely expressed, and is responsive to various hypoxic reactions, especially acute hypoxic reactions, while HIF-2α is related to chronic hypoxic reactions; it is only expressed in some specific tissues. They have their own target genes, but they also have common target genes, such as *VEGF*. HIF-1, but not HIF-2, promotes cellular glucose consumption [80]; HIF-3α is expressed in human kidney and lung epithelial cells. The function of HIF-3 remains to be fully elucidated [81].

The HIF-1 and HIF-2 subunits are similar in DNA binding and dimerization domains, but different in transactivation domains [80]. Therefore, HIF-1α and HIF-2α not only maintain the physiological status of the body with a large number of overlapping and complementary functions [82], but also sometimes show the opposite activity in tumors, and even show specific expression characteristics [83]. The iron absorption by HIF-1α is unnecessary, but HIF-2α plays a key role in maintaining iron homeostasis by directly regulating gene transcription-encoding divalent metal transporter 1 [84]. 

### 4.2. The Role of Hypoxia Environment in the Body

With precision, O_2_ diffuses from plasma to tissue, and then transmits to the inside of cells. The basic purpose of the vascular system is to deliver O_2_, nutrients and to remove carbon dioxide (and other metabolites) from the cells. Therefore, oxygenated blood is distributed to different tissues based on their functions; the physiological O_2_ partial pressure (pO_2_) in the human body is much lower than the atmosphere, ranging from 1% in cartilage and from 1 to 7% in bone marrow to 10–13% in the arteries, lungs and liver [85]. Measurements of pO_2_ in young adult mice yielded the low values of 6.3%, 3.9% and 1.5–2.5 % in the periosteum, cortical bone and marrow [86]. The HIF-1α signal increases glycogen storage, prevents nutrition and energy deficiency during hypoxia, and thus has a beneficial effect on cell survival. Therefore, targeted cell metabolism is an important strategy for bone regeneration and cell therapy [87]. As for bone tissue, microcirculation disturbance causes nutrients not to enter into the bone normally through the Haversian system, which results in bone losing nutrition and leads to OP directly. Also, it has been found that anaerobic metabolism and the accumulation of acid metabolites can lead to the slight acidification of the local microenvironment, while the activation of OC depends on it [86]. Therefore, it is unscientific to study the biology of different cell types in the same O_2_ concentration.

### 4.3. Regulatory Mechanism of Hypoxia

The HIF is constituted of HIF-α and HIF-β. The N-terminal of the α subunit activation region contains an ODD region, which contains a group of specific proline residues, Pro402 and Pro546 in HIF-1α and Pro405 and Pro531 in HIF-2α [70]. HIF-prolyl hydroxylation promotes interaction with the VHL ubiquitin E3 ligase complex and targets HIF-subunits for degradation by the ubiquitin-proteasome pathway. HIF-asparaginyl hydroxylation interferes with the binding of p300/CBP co-activators to the C-terminal activation domain and thus inhibits transcriptional activation, independently effecting protein stability [88]. DFO, CoCl_2_ and DMOG can reduce the activity of PHD and block the degradation of HIF-1α by chelating iron, competing iron binding sites and competing α-ketoglutarate [75].

In the presence of the hypoxia environment or hypoxia simulators (such as DFO, CoCl_2_ and DMOG), the enzyme pathway is inhibited and the level of HIF-α in the cytoplasm increased. After dissociation from Hsp90, HIF-1α enters the nucleus and combines with HIF-1β to form an active HIF-1 complex; with the action of costimulatory factors such as CBP/P300, SRC-1, etc., they bind to the HRE of the target gene, promoting transcription and cause a series of cell responses to hypoxia. The downstream target genes of HIF-1α include *EPO*, *VEGF* and so on [87]. *EPO* gene expression is modulated by HIF-1α. After binding with the EPO receptor, the JAK2-STAT5 pathway and PI3K-AKT pathway are induced, thus enhancing erythropoiesis [89]. Therefore, only when the ambient oxygen concentration is low, or the hypoxia simulator exists, will it activate the downstream target genes and trigger a series of reactions that adapt to hypoxia or changes of the disease’s development. This section is summarized in Figure 1.

Schematic diagrams of the hypoxia pathway. The cellular response to O_2_ is a central process in cells. This process is coordinated by the HIF and its regulators. The HIF induces the expression of proteins controlling glucose metabolism, cell proliferation and vascularization. Several genes involved in cellular differentiation are directly or indirectly regulated by hypoxia. These include EPO, VEGF, etc. The HIF consists of HIF-α and HIF-β. HIF-1α accumulates under hypoxia conditions whereas HIF-β is constitutively expressed. HIF-β is the same as the ARNT, an essential component of HIF protein. In the presence of O_2_, the HIF is targeted for destruction by an E3 ubiquitin ligase containing pVHL. Human pVHL binds to a short HIF-derived peptide when a conserved proline residue at the core of this peptide is hydroxylated. Recently, a factor inhibiting HIF-1α activation, FIH, has been described, representing a further level of HIF regulation.

## 5. Hypoxia Pathway in OP, Osteoblast, Osteoclast and Osteocytes

Osteoporosis is characterized by low bone mass and the destruction of the bone microstructure associated with iron overload; cell dysfunction, including osteoblasts, osteoclasts and osteocytes that sense abnormal changes in response to hypoxia signaling, also occurs. The data show that the VHL-mediated signal transduction of osteochondral precursor cells plays a key role in bone remodeling after birth/adulthood by coupling osteogenesis and angiogenesis [90]. The loss of the *VHL* gene in osteochondral progenitor cells of adult bone could protect mice from aging bone loss; surprisingly, HIF-1 did not affect the vascularization of the skull. In the process of skull formation, activating early HIF-1 regulated the periosteum upward in dense mesenchymal cells [91]. 

The growth and development of normal bone cannot be separated from the existence of hypoxia; the occurrence of OP disease will also cause the abnormality of the hypoxia pathway [92], so the research on the role of the hypoxia pathway in OP and its interaction are essential. The following focuses on the relationship between the hypoxia pathway and OP.

### 5.1. Research Progress and the Regulation between the Hypoxia Pathway and OP

The relationship between hypoxia and OP was first mentioned in 1999. When Wistar rats were pair-fed and their locomotion was limited, the group placed for 4 weeks under hypoxic air showed a reduction in BMD as compared with the control, which suggested that hypoxemia contributed to bone loss [93]. A study showed that prenatal hypoxia delayed the growth of fetal bone [94]. The decrease in blood flow in the lower extremity that might be related to the increase in bone loss rate, the targeted effect of angiogenesis, could promote bone regeneration [95].

Spaceflight led to OP; astronauts exhibited a decreased aerobic exercise capacity, cardiac output and O_2_ diffusional conductance [96]. While in the model of mouse hindlimb disuse, the decrease in O_2_ convection transport could not explain the loss of cortical bone, which was responsive to the mechanical conduction of bone loading and unloading [97]. It was reported that HIF was a new inhibitor of cathepsin B and K [98]. Liu [99] et al. reported that PHD inhibitors DFO and DMOG increased BMD, bone microstructure and bone mechanical strength in OVX rats, and promoted angiogenesis to protect bone loss. Xue [100] et al. proved that DFO decreased the iron content in the bone to relieve bone loss. Zhao [101] et al. suggested that with the knocked out *VHL* gene in OB, the expression of *HIF-1α*, *HIF-2α*, and the *VEGF* gene was up-regulated, and the bone mass and blood vessels in bone marrow were increased in estrogen deficiency-induced bone loss. These findings may help to elucidate the pathophysiology of OP caused by decreased vascular supply [102]. 

The oral administration of 2-methoxyestradiol had protective effects in OVX mice [103]. It was also proved that the knocked-out *PHD2* gene activated the HIF-1α signaling pathway, which led to the decrease in sclerostin (SOST) expression and the enhancement of the response to the Wnt/β-catenin pathway, which finally showed the accumulation of bone mass [104]. Bu Shen Tong Luo decoction as a Chinese traditional medicine increased VEGF expression through the HIF-1α pathway, which significantly reduced the mRNA and protein levels of the calcitonin receptor and cathepsin K, also inhibiting the absorption of the epiphyseal bone of the femoral shaft in OVX rats [105]. In vivo studies confirmed that SAL reduced the concentrations of minerals; ALP improved the bone trabecular microstructure of the distal femur in the OVX rats, which was relative to the activation of the HIF-1α pathway [106]. Isoquercetin was excavated to improve the histological characteristics of OP, which was equivalent to 17β-estradiol by inhibiting the expression of HIF-1α and increasing the VEGF and β-catenin expression, which also down-regulated NF-κB levels [107]. Moreover, GO analysis and KEGG enrichment analysis illustrated that XIANLINGGUBAO prescription played a role in OP treatment through the IL-17 signaling pathway, HIF-1 signaling pathway, insulin resistance, Th-17 signaling pathway [108], etc. Shao [109] et al. explained that the expression of OPG in HIF-1α knocked-out mice was down-regulated, showing a declining trend of bone mass. 

It was also reported that HIF-1α was an osteogenic factor in the formation of woven bone after traumatic loading, but it was an anti-osteogenic factor in the formation of lamellar bone after basal conditions and non-traumatic loading [110]; while the mice lacking HIF-1 in osteocytes were subjected to tibial load, more bone was formed due to the increase in OB activity. However, Riddle [111] et al. suggested that HIF-1 could be used as a negative regulator of bone mechanical transduction to inhibit load-induced bone formation by changing the sensitivity of OB and osteocytes to mechanical signals. Recently, a study proved that HIF-2α was highly expressed in aged bone. The osteoblast-specific or osteoclast-specific conditional knockout of HIF-2α in mice could reverse age-related bone loss [112]. The hypoxia pathway is very closely relatived to OP, and there is much basic research, and some significant results. However, the existing research has not produced effective hypoxia preparations or drugs for prevention and treatment. Therefore, it is necessary to further study their internal relationship in order to get the hypoxia-related preparations for clinical research. The hypoxia pathway is not only directly related to OP, but also to bone-related cells in OP, including OB, OC and osteocytes. Here, a detailed introduction follows. This section is summarized in Figure 2.

Schematic diagrams studying the hypoxia pathway and bone tissue in OP. The HIF has importance in bone remodeling and homeostasis. The knocked-out *HIF-1* gene could lead to low bone mass through modulating OPG expression, while the inactivated *VHL* gene leads to the activation of HIF-1and HIF-2, and gained bone mass. Bu Shen Tong Luo Decotion, 2-methoxyestradiol, SAL, isoquercetin and xianlinggubao prescription activated the HIF-1 pathway, which was useful in OP disease; VEGF, β-catenin, NF-κB, the IL-17 signaling pathway were all part in the process.

### 5.2. Research Progress and the Regulation between Hypoxia Pathway and OB

The reduction of OB production is beneficial to the occurrence of OP. The hypoxia environment is indispensable in the growth and differentiation of OB. OB sensed and responded to changes in the biophysical signals and oxygen supply, which had a profound impact on bone [113]. Lactate enhanced the activity of ALP and induced OCN expression; it also stabilized the expression of HIF-1α in OB culture [114]. Transfer-related gene 1 (such as *MTA1*) was also involved in the growth and differentiation of MC3T3-E1 cells [115]. OB and hypoxia were first studied in 1989. It was reported that hypoxia induced bone resorption by inhibiting mineralization activity and enhancing the production of PGE2 in OB [116]. Hypoxia also inhibited the expression of SOST by antagonizing BMP signals unrelated to VEGF in OB [117]. Hypoxia promoted the expression of BMP-2 through a HIF-1α mechanism in OB activation of the ILK/Akt and mTOR pathways [118]. The HIF pathway in OB could also regulate bone homeostasis through EPO production [119]. If the *HIF-1α* gene was knocked-out in OB, the long bone was significantly thinner and the blood vessels were significantly reduced [120]. ATF4 was a key regulator of VEGF release from the HIF/VEGF axis and bone matrix in OB [121]. OB are derived from mesenchymal stem cells, and there have been some studies on the osteogenic differentiation of mesenchymal stem cells. Costa, V. et al. demonstrated that miR-675-5p might promote hMSC osteogenesis by increasing the HIF-1α response, and also activated Wnt/β-catenin signaling [122]. miR-33a acted as a regulatory factor that drove hypoxia signaling and cytoskeletal recombination in hMSCs osteogenesis. Other signals involved in this process included epithelial-to-mesenchymal transformation (EMT) and epidermal growth factor receptor (EGFR) signaling by regulating the expression of yes associated protein (YAP)/ PDZ binding elements (TAZ) [123]. The co-treatments LIPUS and the miR-31-5p inhibitor abolished the hypoxic responses including angiogenesis and the expression of Rho family proteins. MiR-31-5p was identified as a LIPUS-mechanosensitive miRNA and might be considered a new therapeutic option to promote or abolish hypoxic response and cytoskeletal organization on hMSCs during the bone regeneration process [124].

The hypoxia pathway is not only essential to the normal development of OB but plays an important role in the development of OB in OP disease. The study of OB and hypoxia in OP was first undertaken in 2007 [125]. Bone regeneration involves a series of coordinated events, including the recruitment of mesenchymal stem cells, the induction of immune response, inflammatory activity and vascular growth. Genetic methods showed that the overexpression of HIF-α in mouse OB by destroying VHL led to a significant increase in angiogenesis and osteogenesis [126]. Hypoxia or CoCl_2_ exposure significantly increased the level of *HIF-1α* protein and *VEGF* gene expression, but owned different mechanisms [127]. DFO was an iron chelator that already currently licensed promoting angiogenesis and bone formation, and also relieved the symptoms of osteoporosis [128]. DMOG [129] promoted the proliferation and differentiation of OB by inhibiting PHD activity to induce the HIF pathway. HIF-1α cooperated with osterix, a specific transcription factor in OB, to inhibit the Wnt pathway [130]. The overexpression of miR-135-5p promoted the differentiation and calcification of MC3T3-E1 cells by targeting the HIF-1α inhibitor (HIF-1AN) [131], while the knocked-out *miR-135-5p* gene produced the opposite result. OASIS/CREB3L1 is an ER-resident transcription factor that plays an important role in OB differentiation. It was suggested that OASIS could affect the expression of the HIF-1α target gene and participated in angiogenesis during bone development [132]. The partial lack of HIF-1 led to the decrease in chondrocytes and OB apoptosis, which allowed the development of larger and harder calluses. Thus, the inhibition of apoptosis was a promising target for the development of new therapies to accelerate bone regeneration [133]. In addition, miR-210 stimulated the differentiation of OB by stimulating HIF-1α and VEGF [134] in OP. miR-1-3p enhanced the differentiation of MC3T3-E1 cells by targeting and interacting with HIF-1AN directly [135]. Other scholars proved that HIF-2 was an inhibitor of OB growth and bone accumulation [136]. Therefore, the inhibition of HIF-2 may be a new and major direction in the treatment of low bone mass diseases.

The hypoxia pathway can regulate the growth and differentiation of OB by combining multiple pathways. In OP, the growth and development of OB is closed to the regulation of the hypoxia pathway, but there are different opinions on the relationship between OB and the hypoxia pathway, mainly focused on cell type, treatment time, treatment measures, O_2_ concentration, etc.

This section is summarized in Figure 3.

Schematic diagrams studying the hypoxia pathway and osteoblasts. HIF-1 made sense of cell growth, differentiation and calcification, angiogenesis, bone accumulation and generation by Wnt/β-catenin, Notch, BMP, ILK/AKT, mTOR and VEGF pathways. HIF-2, miR-135-5p and dexamrthasone had negative effects on bone homeostasis, while miR497-195, miR-210, miR-1-3p played a coupling role in angiogenesis and bone formation. DFO, CoCl_2_, DMOG activated the expression of HIF-1 by hindering PHD activity, and the knocked-out *PHD2* gene also owned the same effects.

### 5.3. Research Progress and the Regulation between the Hypoxia Pathway and OC

OC are multinucleated hematopoietic cells that can absorb bone and are associated with OB in a dynamic balance during bone modeling and remodeling. However, if the balance is broken, OP will occur. The increased production of OC is a common feature of pathological bone loss.

The relation between OC and hypoxia was first studied in 2003. Short term cell cultures isolated from rat bone showed that hypoxia did not change the absorption activity of mature OC but reduced their survival rate or adhesion [137]. In 2011, it was reported that icariin inhibited OC formation by inhibiting p38 and the JNK pathway in OP [138]. The expression of fibroblast growth factor 11 (FGF11) in OC stimulated by hypoxia was related to the co-localization of microtubule-associated α-tubulin [139]. In the hypoxia microenvironment, the interaction of HIF and the adenosine A2B receptor could enhance the glycolysis and mitochondrial metabolism of OC, thus promoting the increase in bone resorption [140]. However, the data showed that the HIF was mainly a regulator of OC-mediated bone resorption, which had little effect on OC differentiation [141]. The activation of OC in postmenopausal OP due to estrogen deficiency required the involvement of HIF-1α [103]. It reported that HIF inhibitors are potential therapeutic drugs for targeting bone resorption [142]. MicroRNA was also involved in the regulation of OC by the hypoxia pathway [143]. SERM [144] have been developed to inhibit the accumulation of the HIF-1α protein in mouse OC for OP treatment [145]. The PHD inhibitor FG4592 could mediate the homeostasis between osteoblasts and osteoclasts in vitro and proved to have a positive regulatory effect on bone remodeling, making it a potential drug for osteoporosis in clinical trials [146]. 

The activation and differentiation of OC was closely related to the hypoxia pathway, and this chapter also put forward the hypoxia inhibitor to inhibit the generation and increasing amounts of OC. However, the different treatment methods and research plans of OC lead to different responses to hypoxia concentration. Moreover, OP is the result of the joint action of OB, OC and osteocytes. How to find effective treatment drugs from OC is the difficult problem and direction of future research.

### 5.4. Research Progress and the Regulation between the Hypoxia Pathway and Osteocytes

Osteocytes are the most abundant cells in bone tissue. In fact, the osteocytes encapsulated in the mineralized bone matrix are multifunctional cells that play a key role in the regulation of bone and mineral homeostasis. In addition to being endocrine cells and regulating phosphate homeostasis, these cells control bone remodeling by regulating OC and OB as well as controlling hematopoiesis and myeloid cells via multiple mechanisms. They are mechanical sensory cells, coordinate the adaptive response of bone to mechanical load, and also act as managers of the bone calcium pool [147]. However, there is little research on the hypoxia pathway in osteocytes. In early 2001, it was found that acute discontinuation (1–5 days) resulted in a significantly higher percentage of HIF-1-positive bone cells than normal bone, and this response was consistent with the cortex. In addition, acute hypoxia (4–12 h of 2% O_2_) resulted in the up-regulation of HIF-1 protein expression in MLO-Y4 cells by 2.1–3.7 times compared with the cells cultured in parallel under normal O_2_ conditions [148]. Bone loss might be due partly to the death of osteocytes in OP [149]. Glucocorticoids inhibited the angiogenesis of fetal metatarsal and the transcription of HIF-1, as well as the production of VEGF in osteocytes [150]. HIF-1α promoted the expression of RANKL by activating the JAK2/STAT3 pathway of MLO-Y4 cells [151]. In vitro studies also showed that the chemical method led to the hypoxia-activated high expression of HIF-1α, which induced MLO-Y4 cell apoptosis, and promoted osteoclast production mediated by osteocytes. At the same time, the JNK/Caspase-3 signaling pathway was involved in it [152].

It was studied that the VHL/HIF-α pathway in mature OB/osteocytes played a key role in the network of osteocytes/tubules. The loss or inactivation of the *VHL* gene led to the apoptosis of cells. The accumulation of β-catenin in bone marrow OB/bone progenitor cells changed the morphology/function of osteocytes and the network of osteocytes/tubules [153]. While it showed that the absence of VHL in osteocytes resulted in a high bone mass phenotype, HIF-1a was not essential in osteocytes; the function of VHL/HIF-α of osteocytes maintains bone microstructure and influences hematopoiesis through the Wnt pathway [154].

Osteocytes have become one of the longest surviving cells in vivo. Therefore, viability and survival rates are key to ensure the optimal function of the osteocyte network. In addition to OC and OB, osteocytes should also be included in new strategies for the prevention and treatment of OP, and guidance provided on how to research OP drugs on osteocytes, which needs further exploration. However, the potential risk of gene knock-out should be paid more attention, such as the activation of the proto-oncogene, the inhibition of the tumor suppressor gene, the recurrence and metastasis of cancer [155].

The content summary of the above two parts is shown in Figure 4.

Schematic diagrams studying the hypoxia pathway, osteoclasts and osteocytes. HIF-1 is mainly involved in the bone resorption in glycolysis and the mitochondrial metabolism of osteoclasts. SERM inhibited the accumulation of HIF-1α protein in mouse osteoclasts. Icariin inhibited OC formation by inhibiting p38 and the JNK pathway in OP. MicroRNA and VEGF were also involved in the regulation of osteoclasts by the hypoxia pathway. In osteocytes, the VHL/HIF-α pathway in mature osteoblasts/osteocytes played an essential role in the network of osteocytes/tubules; the *VHL* gene was inactivated and led to cell apoptosis. Glucocorticoids inhibited the angiogenesis and transcription of HIF-1 in osteocytes. HIF-1α promoted the expression of RANKL by activating the JAK2/STAT3 pathway.

## 6. Application of O_2_, Hypoxia Pathway in OP/Iron Overload Diseases

### 6.1. Hyperbaric Oxygen Chamber Therapy and OP

There was no adverse effect of hyperbaric oxygen (HBOT) therapy (100% O_2_; 2.4 Atmospheres Absolute for 90 min) on bone homeostasis in 20 patients aged 35–82 years with complications caused by radiotherapy or chronic anal fissure; the data suggested that serum HIF-1α levels were increased in eight patients, while serum OPG, RANKL, sclerostin, DKK1 levels were not significantly affected by HBOT. Hyperbaric oxygen chamber therapy could increase the collagen deposition, restore bone progenitor cells (especially cell activity) and bone microcirculation, improve the injury of ischemia hypoxia, extend the degradation period, improve the hardness and flexibility of bone [156]. Now, there are many cases and basic studies of the rehabilitation treatment of OP symptom patients using the HBOT chamber. Twenty postmenopausal women (38–64 years old) were treated 15 times with HBOT at 1.5 Atmospheres Absolute and 100% O_2_. The BMD results showed a significant 18.5% increase in lumbar spasms in 19 of 20 patients, a significant reduction in pain in 17 patients, and clinical improvement in lumbar range of motion in 15 patients. No side effects were observed in the treated patients [157]. Studies seem to indicate that HBOT has been successfully used to relieve pain in patients with transient osteoporosis of the hip (TOH) who did not respond to other conservative treatments [158]. Exposure to mild hyperbaric oxygen (40% O_2_) partially protected osteoporosis caused by hind limb unloading by inhibiting osteoclasts, enhancing bone formation and reducing Sost expression [159]. A diagnosis of transient osteoporosis of the hip (TOH) was made and the patient received a total of 30 sessions of hyperbaric oxygen (HBO_2_) at 2.4 ATA; the pain was gradually relieved and the patient became asymptomatic after one month together with a complete recovery of the range of motion of the hip [160]. Five cases of TOH treated with HBO_2_ were clinically evaluated. Multiple HBO_2_ treatments had the possibility of contributing to recovery acceleration in patients with TOH [161]. Metabolic disorders in obese and non-obese aging rats (NDD and HFDD) were observed by measuring the bone microstructure and biomechanical strength of metabolic parameters, HBOT restored bone remodeling and bone structure/strength in the obese aging rats, while HBOT improved bone balance abnormalities in the non-obese aging rats, which is a potential intervention to reduce the risk of osteoporosis and fracture in the elderly [162].

In conclusion, the hyperbaric oxygen chamber can improve vascular microcirculation, promote bone formation, inhibit bone absorption, regulate and alleviate bone metabolism abnormalities in patients with osteoporosis so as to improve bone mechanical properties and metabolism and prevent and treat the occurrence and development of osteoporosis.

### 6.2. Mechanical Stress, Hypoxia and Bone Remodeling

Mechanical stress plays an important role in bone remodeling. Mice with HIF-1α deleted in the osteoblast lineage (ΔHIF-1α) had significantly lower woven bone yield than wild-type mice at 7 days after injury loading. The results indicated that HIF-1α is a contributing factor of woven bone formation after mechanical loading at the injury site [110]. The exogenous VEGF dose on bone regeneration under different mechanical stimulation was studied by a 3D mechano-chemical model of bone regeneration and showed that appropriate mechanical stimulation could improve the effect of VEGF on bone regeneration [163]. Cyclic mechanical stretch increased ALP activity and the expression of HIF-1α and TWIST in BMSCs. The HIF-1α-TWIST signaling pathway inhibited the cyclic mechanical stretch-induced osteogenic differentiation of BMSCs [164]. Demethyltransferase fat mass and obesity-associated (FTO) promoted the expression of HIF-1α and osteogenic differentiation under mechanical stress. This finding might facilitate the clinical application and the mechanism research of mechanical stress-induced osteogenesis. [165]. Chondrocytes were subjected to 20% tensile stress under hypoxic (5% O_2_) conditions for 24 h. HIF-1α and aggrecan expression were significantly enhanced. The nuclear translocation of HIF-1α was enhanced by mechanical stress under hypoxia [166]. JING, X. et al. indicated that cyclic mechanical stress promoted HIF-1α stabilization and YAP was involved in mechanical stress-induced HIF-1α up-regulation [167]. Skeletal muscle microvascular endothelial cells exposed to 10% stretch in vitro showed an elevation in HIF-1α and HIF-2α mRNA expression, which was preceded by an increase in HIF-binding activity. Conversely, HIF-1α and HIF-2α mRNA were reduced significantly, and HIF-α proteins were undetectable after 24 h exposure to elevated shear stress. The results illustrated that the activation of HIF-1α and HIF-2α contributed significantly to stretch but not to shear-stress-induced capillary growth [168]. The mechanical microenvironment of stem cells has a huge role in its differentiation, which consists of various factors, including the extracellular matrix and topology, substrate stiffness, shear stress, hydrostatic pressure, tension and microgravity. Nuclear factor-kB, the nicotinic acetylcholine receptor, the piezoelectric mechanosensitive ion channel and HIF-1α are involved in the mechanical modulation process [169].

### 6.3. HIF-Related Drugs and Iron Metabolism Diseases

Iron is one of the essential nutrients in organisms. Iron is involved in many important physiological processes, such as oxygen transport, electron transport, DNA synthesis and so on. Under normal physiological conditions, iron metabolism homeostasis is mainly maintained by complex and sophisticated regulatory mechanisms at systemic and cellular levels. Systemic iron metabolism mainly realizes iron homeostasis through the absorption and efflux of iron in the intestine, blood transportation, tissue utilization, the storage of the liver and the phagocytic cycle of macrophages [170]. Iron metabolism mainly maintains iron homeostasis through iron uptake (transferrin receptor, TFR1; divalentmetal transporter 1, DMT1 mediated), storage (ferritin light chain, FTL; ferritin heavy chain, FTH1 mediated), utilization and excretion (ferroportin, FPN is the only known factor) [171]. O_2_ is closely related to iron homeostasis; iron deficiency can lead to anemia. In contrast, excessive free iron not only stimulates the formation of ROS, but also leads to tissue damage. Aberrant hypoxia is a common basic pathological process in many diseases, and iron is indispensable in normal human activities. It has been found that HIF-2α mediates the up-regulation of intestinal ferritin to increase the adaptability of iron-deficient mice, and FPN targeted therapies might be developed for patients with iron-related diseases [172]. The EPO expression is also influenced by hepcidin and glutamate secretion [173], and it has been emphasized that the ferritin/TFR axis regulates iron absorption with the regulation of intestinal HIF-2 [174]. 

Hypoxia has effects on iron transport, iron oxidation, heme catabolism/iron recycling and iron uptake/availability, especially HIF-2 but also HIF-1-influenced iron homeostasis. It was reported that HIF-2 mRNA was increased in an iron overload model associated with the rise of iron-absorbing proteins, such as transferrin receptors (TfR), ferroprotein (FPN), divalent metal transporter (DMT1), duodenal cytochrome b (DcytB), hepcidin, heme oxygenase-1 (HO-1), ceruloplasmin, EPO, etc., but the expression of hepcidin was decreased. Therefore, HIF-2 inhibition may be a good measure to fight iron overload disease [174,175,176]. In macrophages, transcription factor NRF2 regulated the expression of genes for hemoglobin decomposition and iron metabolism. NRF2 was considered to be a molecular sensor of iron, inducing oxidative stress and regulating iron levels throughout the body. Therefore, NRF2 and HIFs played key roles in the redox regulation of iron homeostasis in the system. HIF-2α was also regulated by the IRP/IRE regulatory system for translation. The regulation of HIF-2α stability by the IRP/IRE system was considered as another potential drug target [177]. A clinical HIF-2α inhibitor PT2385 drug treated iron overload in a HAMP^Δliv^ mouse model [178]; the study certified that PT2385 could reverse the iron accumulation in multiple tissues in this model. The low or no expression of FPN, DMT1 and DcytB was found in the mice, indicating that the hypoxia preparations had important research and guidance value in iron metabolism disorder diseases.

All in all, studying iron metabolism in the hypoxia state can explain the pathogenesis and development of metabolic diseases from another perspective, which provides a new clue and strategy in the field of OP, and also in clinical research.

The summary flow chart of applications and prospects for hypoxia in OP. In a word, the regulation of mechanical stress and the hyperbaric oxygen chamber can improve the hypoxia state in OP, and the treatment of OP can be achieved by regulating iron metabolism disorder in pathological bones. On the basis of existing HIF-2 inhibitors (in an iron overload disease mouse model) and HIF-1 activators (in an OVX/mouse model), the experimental exploration of OP can also be carried out, so as to make breakthroughs and new discoveries in hypoxia preparations and find targets or new compounds.

## 7. Conclusions

This paper focuses on the description of hypoxia pathways in OP and bone-related cells (OB, OC, osteocytes). Based on the complexity of diseases and the importance of the hypoxia pathway, we realize that the mechanism of the hypoxia pathway in OP disease needs to be further elucidated and clarified. The alleviating/improving effects of HBOT chamber equipment on the symptoms of OP were introduced briefly, as well as mechanical stimulation to activate the hypoxia pathway in response to changes in bone-related cells, and the basic research on the treatment of the mouse iron overload model by improving iron metabolism with hypoxic-related drugs. It provides a new idea for the clinical application of hypoxic-related small molecules (HIF-1 activators or its mimics, HIF-2 inhibitors, PHD inhibitors) in the treatment of OP (iron accumulation). In a word, the hypoxia pathway is of great significance in further demonstrating the occurrence, onset process and more mechanisms of OP; we can explore new directions and plans for treatment based on it (See in Figure 5).

## Figures and Tables

**Figure 1 ijerph-20-03129-f001:**
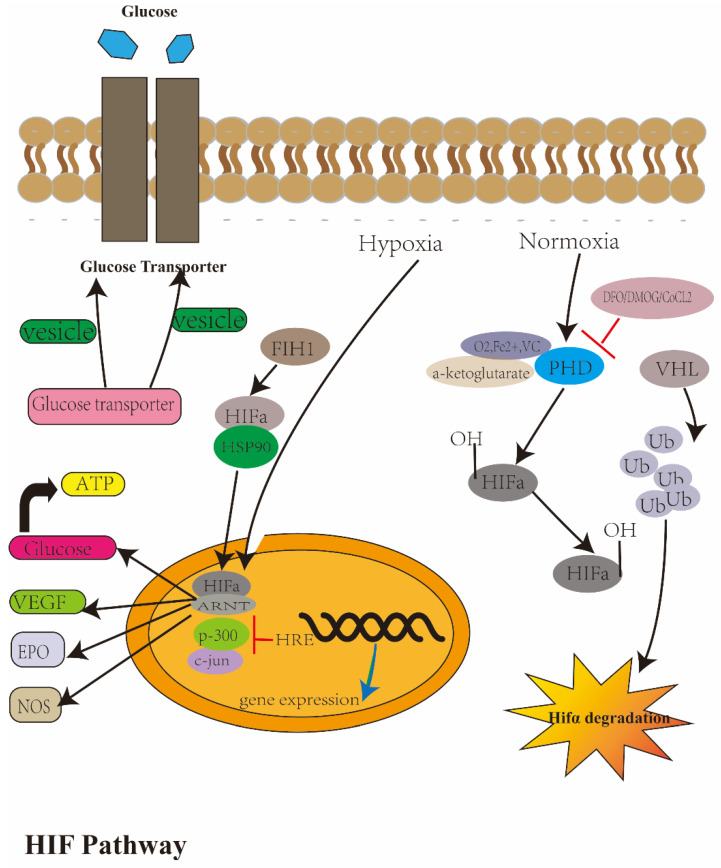
The introduction of the hypoxia pathway.

**Figure 2 ijerph-20-03129-f002:**
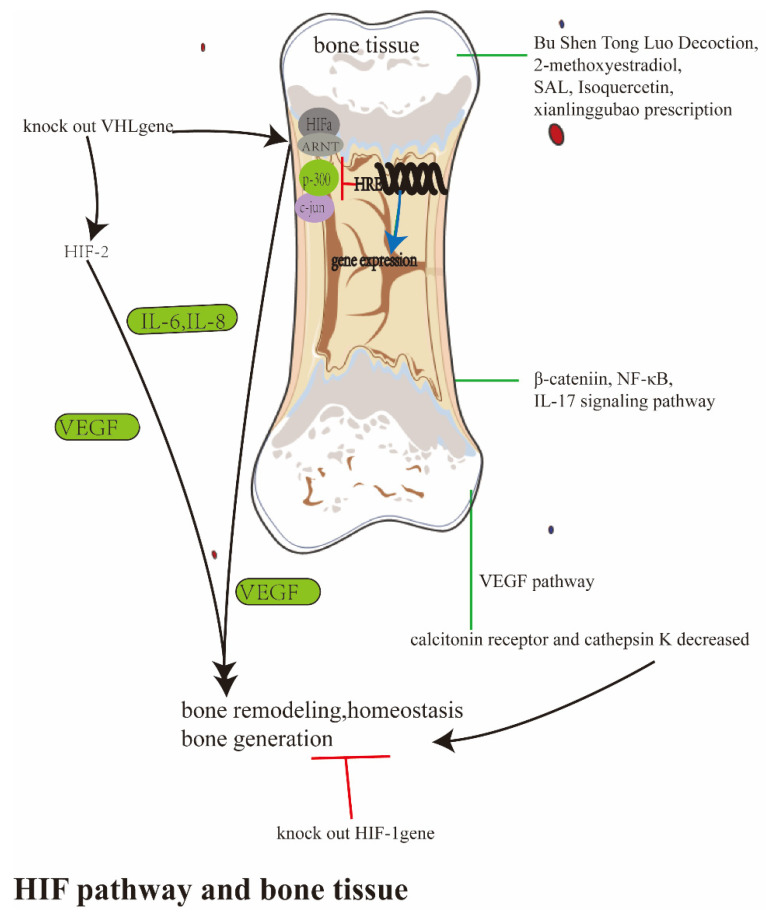
The summary of study between the hypoxia pathway and bone tissue in OP.

**Figure 3 ijerph-20-03129-f003:**
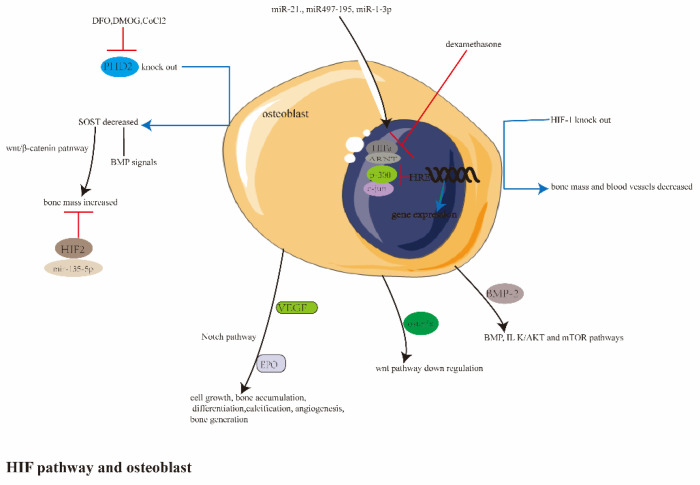
The summary of study between the hypoxia pathway and osteoblasts.

**Figure 4 ijerph-20-03129-f004:**
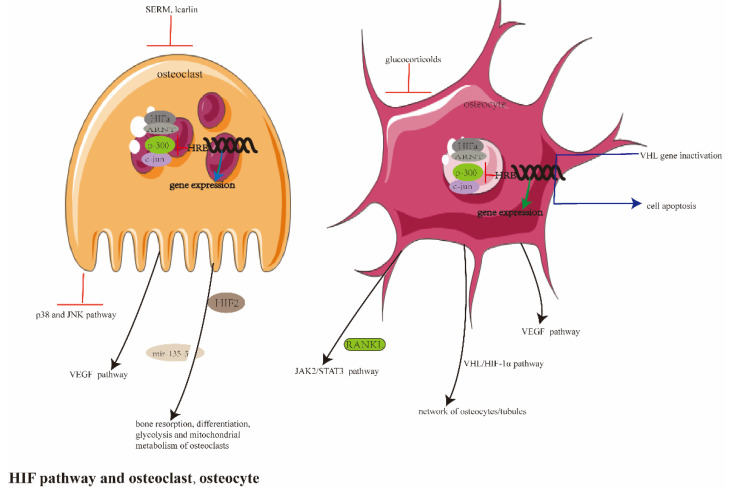
The summary of study between the hypoxia pathway and osteolasts, osteocytes.

**Figure 5 ijerph-20-03129-f005:**
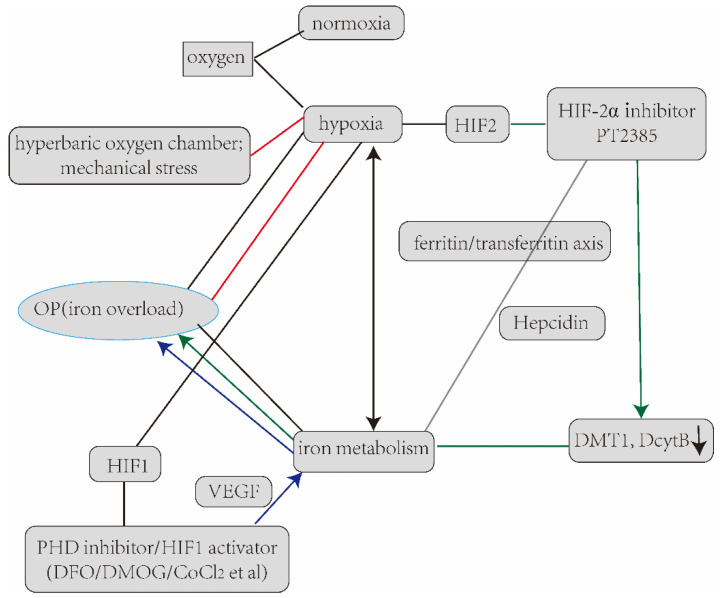
The summary of articles and prospects for hypoxia in OP.

## Data Availability

Data sharing is not applicable to this article as no datasets were generated or analyzed during the current study.

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
