# Peer review of "Hypoxia Pathway in Osteoporosis: Laboratory Data for Clinical Prospects"

_ijerph, 2023, doi:10.3390/ijerph20043129_

Round 1

Reviewer 1 Report

This review summarizes the relationship and regulation between hypoxia pathway and osteoporosis (also include osteoblasts, osteoclasts, osteocytes) by arranging the references on the latest research progress, introduces briefly the application of hyperbaric oxygen therapy in osteoporosis symptoms, mechanical stimulation induces skeletal response to hypoxic signal activation, hypoxic-related drugs used in iron accumulation/osteoporosis model study, also puts forward to the prospects of future researches.

In this manuscript Wang et al. summarizes the relationship/regulation between hypoxia pathway and osteoporosis, introducing briefly the application of hyperbaric oxygen therapy in osteoporosis symptoms, mechanical stimulation induces skeletal response to hypoxic signal activation, hypoxic-related drugs used in iron accumulation/osteoporosis model study in order to improve the basis and knowledges for future specific researche

In particular, to further improve the quality of the manuscript, the authors should discuss better the following points:

             The review evaluated a new aspect of OP disease, but in the last years Bellavia D. and colleagues produced new studies about the role of different miRNAs and their regulation in OP. It is better from me to add these studies in the references section.

             To improve the quality of the manuscript, I suggest adding the new evidences about the role of miRNAs in the specific regulation of Hypoxia signaling during hMSCs osteoblast differentiation or in bone disease (line 51);

             I suggest adding the definition of the acronyms VHL (line 112), PHD (line 122), VC (line 123), EPO (line 181). It is better to insert it in the text when citated or described for the first time in the manuscript;

             To check the format of the text in lines: 152,122,257 to 260, 473,475; After to add the definition of EPO in the line 181, I suggest eliminating it in the line 192.

            To clarify the meaning of the sentences reported in lines: 163-164;

             For all figures reported in the text, I suggest checking the quality of images and the orientation of the squares or circles images used in each picture.;

             To uniform the way in which you write HIF-1a and HIF-2a they are reported in a many different ways in the text;

             The sentences reported in lines 292 is unclear and not supported by good citations: “Hypoxia pathway is not only essential to the normal development of OB”. In the last years, many in vitro and ex vivo evidence are developed about the role of hypoxia in bone regeneration and bone disease also underlining the involvement of different miRNAs and lncRNA in HIF-1 a regulation. It is better to improve the references about the Hypoxia and miRNAs regulation (miR675-5p or miR33a-5p or miR31-5p);

             It was probably lose a part of text of the title reported in paragraph 6.1;

             To control the text of in the title reported in paragraph 6.2.

Author Response

We thank the reviewer for commenting our manuscript carefully. Please see the attachment about comments response.

Reviewer 2 Report

This paper presents a summary of some of the known information currently in the literature regarding hypoxia and osteoporosis, but lacks the rigor of a systematic review, rendering the literature reported and content selection susceptible to author bias. My other comments are:

1) Information regarding osteoporosis, and bone remodeling are mentioned on a superficial level and requires a detailed discussion section which would provide a critical review of findings expected of a journal paper.

2) Add the full form of HIF in the abstract

3) Include a detailed explanation and a few more literature under the sections “Osteoporosis” and “Bone Remodeling”.

4) In my opinion the authors should improve the document. Presenting the methodology of review and the data base sources.

Author Response

(The authors gave the same response as above.)

Round 2

Reviewer 1 Report

The authors answered the questions clearly and well descriptively and positively improved the manuscript. I recommend only a small revision of the language. In my opinion, it can be accepted for publication in this journal